# ALIGNING NEWS AND PRICES: A CROSS-MODAL LLM-ENHANCED TRANSFORMER DRL FRAMEWORK FOR VOLATILITY-ADAPTIVE STOCK TRADING

## ABSTRACT

While Deep Reinforcement Learning (DRL) has shown promise for stock trading, its practical application is constrained by critical gaps that undermine performance in real-world volatile markets, most notably during events like the 2020 COVID-19 market crash. Specifically, existing DRL methods fail to capitalize on textual financial news (a key leading indicator of market sentiment), struggle to model multi-scale temporal dynamics, and lack robustness to extreme volatility, leaving them unable to adapt to sudden shifts in market fundamentals. To address these limitations, we propose a volatility-adaptive, multimodal DRL framework for stock trading integrating pre-trained Large Language Models (LLMs), Transformers, and the Soft Actor-Critic (SAC) algorithm. The framework first uses an LLM-driven module to extract sentiment and event features from financial news, maps price dynamics into the LLM's semantic space via a multi-head attention reprogramming layer, and fuses these modalities via cross-attention to capture intrinsic news-price interdependencies. To enhance state representation, a Transformer encoder models short/long-term news sentiment trends, price fluctuations, and inter-stock correlations, and merges these heterogeneous features into a compact, unified state via multi-head attention. Finally, we incorporate gradient feedback from SAC's critic network to the Transformer, enabling end-to-end optimization of feature learning and trading policy. Empirical evaluations on NASDAQ-100 data show our framework outperforms existing DRL methods in multi-stock trading, while surpassing Transformer-based methods in single-stock prediction, with ablations confirming core modules drive performance gains.

## 1 INTRODUCTION

In the stock market, the primary goal of financial trading is to profit from buying and selling financial assets at advantageous prices to increase asset value. This process also involves asset allocation and risk management, which refers to the rational diversification of assets to reduce the risk in highly dynamic market environments and ensure the relative stability and security of assets. While Deep Reinforcement Learning (DRL) has driven notable advances in stock trading by integrating deep learning (DL) and reinforcement learning (RL) (Jiang et al., 2017; Li et al., 2019; Liu et al., 2018; 2021a). The former, including neural networks (LeCun et al., 1989; Vaswani, 2017; Graves & Graves, 2012), with their strength in parsing high-dimensional data (e.g., price time series, technical indicators), excel at identifying latent patterns in historical market information (Jiang et al., 2017; Lien et al., 2023; Lucarelli & Borrotti, 2020; Morales & Zaragoza, 2012; Sahu et al., 2023); The latter (Meng & Khushi, 2019; An et al., 2022), on the other hand, enable trading agents to iteratively learn optimal decision-making policies through continuous interaction with dynamic market environments.

Yet, their synergy remains hamstrung by critical limitations that severely undermine real-world effectiveness, particularly in navigating complex and volatile market conditions, like the 2020 COVID-19 market crash or the 2018 US-China trade friction. First, conventional DRL models, such as Deep DPG (DDPG) (Lillicrap, 2015), Proximal Policy Optimization (PPO) (Park et al., 2024; Schulman et al., 2017), Advantage Actor-Critic (A2C) and SAC (Haarnoja et al., 2018), can not fully utilize multimodal data information. They rely almost exclusively on structured price sequences, over-

looking the pivotal role of unstructured textual data (e.g., financial news, policy announcements) in shaping market sentiment and expectations. Text often precedes tangible price reactions (Guo et al., 2023; Trichilli & Boujelbène Abbes, 2023; Navarro et al., 2023; Benhamou et al., 2021). For instance, during unforeseen events, public opinion dissemination can anticipate price shifts by days or weeks. While Large Language Models (LLMs) tailored to finance, such as FinBERT (Liu et al., 2021b; Huang et al., 2023; Zhao et al., 2021) and FinGPT (Liu et al., 2023) have demonstrated promise in extracting sentiment and event-driven features from text, their integration into DRL remains fragmented. Existing methods (e.g., (Ding et al., 2023)) fail to resolve the misalignment between text semantic features and price-derived representations, limiting the collaborative potential of multimodal data (Cao et al., 2025).

Second, most DRL systems struggle to reconcile multi-scale temporal dynamics simultaneously. Market behavior exhibits nested patterns, from short-term daily volatility to long-term weekly/-monthly trends, yet traditional DRL approaches struggle to handle these scales (Yang et al., 2020). Some state-of-the-art approaches that integrate Transformers, which are well-suited for time-series modeling (Vaswani, 2017)) into DRL fall short in the fusion of market temporal dynamics information. For instance, , while TACR (Lee & Moon, 2023), based on Decision Transformers (Chen et al., 2021) and SAC, focuses on long-term asset allocation dependencies without addressing multi-scale feature fusion. This gap leaves DRL agents unable to fully contextualize price movements, leading to suboptimal decision-making.

Third, existing DRL algorithms often fail to adjust policies dynamically during systemic shocks. For example, during market crashes, these models may suffer from catastrophic losses due to rigid feature representations and disjointed optimization of feature learning and policy execution. While frameworks like FinRL (Li et al., 2021; Liu et al., 2021a) have standardized DRL for finance and proposed adaptive variants (e.g., adaptive DDPG (Li et al., 2019)), they still lack mechanisms to unify multimodal semantics features and volatility awareness into a coherent decision-making process. StockFormer (Gao et al., 2023) uses Transformer branches to capture long/short-term price trends and inter-stock correlations but lacks explicit modeling of volatility dynamics, thereby leading to limited performance in real-world volatile markets.

To address these limitations, this paper proposes a volatility-adaptive multimodal DRL framework that integrates pre-trained LLMs, Transformers, and SAC. The key contributions are:

- A multimodal LLM-driven module that bridges text and price data: We use pre-trained LLMs (e.g., BERT, GPT-2 Radford et al. (2019)) to extract fine-grained sentiment and event features from financial news, and a multi-head attention reprogramming layer (inspired by Time-LLM (Jin et al., 2023))to align price time-series (OCHLV + technical indicators) with the LLM's semantic space, enabling seamless cross-modal fusion via cross-attention

- A Transformer-based feature extractor for multi-scale dynamics: This module models short/long-term news sentiment trends, price fluctuations, and inter-stock correlations, fusing these into a compact state representation via multi-head attention. Critically, we enable gradient feedback from SAC's critic to the Transformer, achieving end-to-end optimization of feature learning and trading policy.

- Empirical validation of superiority in real-world datasets, such as NASDAQ-100 data, our framework outperforms state-of-the-art DRL methods (e.g., SAC, PPO, StockFormer) in multi-stock trading and surpasses Transformer-based and LLM-only models (e.g., Autoformer Wu et al. (2021), GPT-2) in single-stock price prediction. Ablation studies confirm that unifying multimodal semantics and multi-scale temporal dynamics is the core driver of enhanced market volatility adaptability.

## 2 METHODOLOGY

Addressed the limitations of existing DRL approaches in stock trading, our method integrates pre-trained LLMs and Transformers into a SAC-based multimodal DRL framework (Figure 1). In the observation extraction module, LLMs conduct fine-grained semantic parsing and sentiment quantification on financial news, while a multi-head attention reprogramming layer maps OCHLV data (Open, Close, High, Low and Volume) and technical indicators into an LLM-compatible space for seamless multimodal fusion. The state generation module employs a Transformer-based predictive

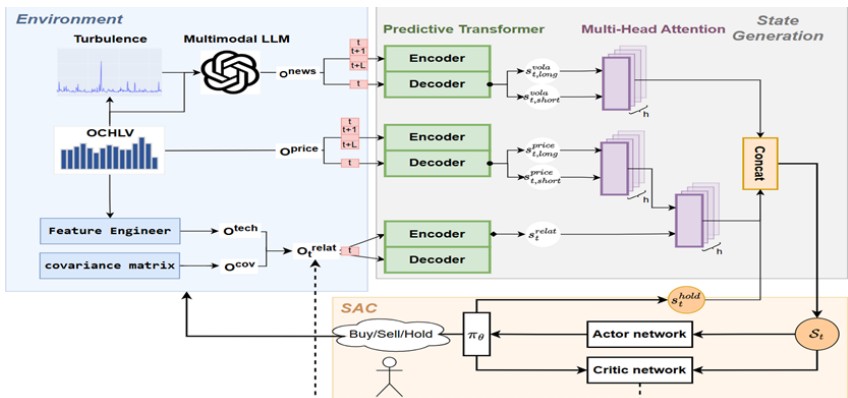

Figure 1: The proposed framework integrating pre-trained LLMs, Transformers into SAC-based DRL for stock trading, with observation extraction, state generation and critic gradient feedback.

model to capture short/long-term news trends, price dynamics, and inter-stock correlations, fusing them via multi-head attention into a unified state. This state is fed into SAC for adaptive trading, where critic gradients propagate back to enhance joint training of predictive features encoding and policy learning, strengthening the agent's ability to leverage multimodal insights for robust decision-making.

## 2.1 MULTIMODAL LLM

To enable cross-modal alignment (for observation extraction) and enrich the state space, we design a multimodal LLM designed to integrate financial news and price data, perform predictive tasks and capture latent market signals. It learns latent representations by fusing price time-series ($P = \{p^{\text{open}}, p^{\text{close}}, ..., p^{\text{volume}}\}$) and news ($p^{\text{news}}$),and the model minimizes supervised prediction loss by learning latent representations $h^{\text{news}}$ from multimodal data $P$, enabling it to accurately perceive market conditions. Specifically, using a sliding window of size $D$ on the multimodal sequence data $P_{n,t-z:t}$, we predict the stock's closing price at the next time point $p^{\text{close}}_{n,t+1}$, formulating the following objective:

$$\min_{\theta} \mathbb{E}[\mathcal{L}(f(P_{n,t-z:t}), p^{\text{close}}_{n,t+1})] \tag{1}$$

where $f$ is the time-series LLM model parameterized by $\theta$, and $\mathcal{L}$ denotes the loss function. Through this process, the model autonomously learns potential market trends and events driven by multi-modal information, enriching downstream analysis and decision-making, with three core components including Price Encoder, News Encoder and Fusion Module, as depicted in Figure 2.

**a) Price Encoder:** To convert price time-series into LLM-compatible tokens, we first preprocess OCHLV data via standardization (to normalize scales) and and partition the sequence into fixed-length patches (e.g., chunks of $k$ consecutive time steps). Each patch is embedded into a dense vector via a linear projection, generating chunked patch embeddings $X_{\text{price}}$ that explicitly preserve temporal structure for capturing sequential price dynamics.

**b) News Encoder:** The News Encoder leverages pre-trained LLMs to process financial text, encoding news into contextual embeddings ($h^{\text{news}}$) capturing semantic nuances and market sentiment. To align these embeddings with stock prediction tasks (e.g., closing price forecasting), we employ task-specific prompt engineering (Figure 3a) and explicitly specify input features (historical prices, lag terms, news content) and guide the LLM to prioritize market-relevant patterns,ensuring semantic representations align with downstream trading objectives.

**c) Reprogramming layer:** As shown in Figure 3b, the reprogramming layer aligns price patch embeddings $X_{price}$ with the LLM's semantic space, which is critical for enabling effective fusion of price and news tokens. By remapping time-series inputs to an LLM-compatible space using text prototypes, this layer supports universal time-series modeling without modifying the LLM backbone. Using a token embedding matrix $E \in \mathbb{R}^{V \times D}$ (where $V$ is vocabulary size and $D$ is hidden

dimension), multi-head attention projects price features into the LLM's representation space:

$$Q_h = X_{\text{price}} W_h^Q$$
$$K_h = X_{\text{price}} W_h^K \tag{2}$$
$$V_h = X_{\text{price}} W_h^V$$

This projection yields price embeddings $Z_{\text{price}}$ compatible with news-derived tokens:

$$Z_{\text{price}} = \text{Concat}(Z_1, \ldots, Z_h) W^O \tag{3}$$

$$Z_h = \text{SoftMax}\left(\frac{Q_h K_h^{\top}}{\sqrt{d}}\right) V_h \tag{4}$$

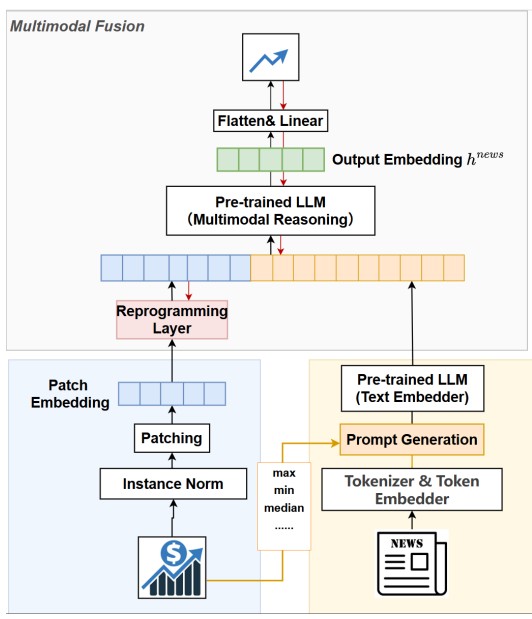

Figure 2: Multimodal LLM framework: Price Encoder (lower left), News Encoder (lower right), and Fusion Module (upper) for cross-modal alignment.

**d) Multimodal Fusion** The multimodal fusion module leverages cross-attention mechanisms to enable token-level interaction between aligned price embeddings $Z_{\text{price}}$ (from the Price Encoder)and news embeddings $Z_{\text{news}}$ (from the News Encoder) by enabling token-level interaction, which is critical to capture intrinsic news-price interdependencies:

$$Z_{price-news} = \text{Attention}(Q_{news}, K_{price}, V_{price})$$
$$Z_{news-price} = \text{Attention}(Q_{pirc}, K_{news}, V_{news}) \tag{5}$$

where the query (Q), key (K), and value (V) matrices are defined as:

- $Q_{news} = Z_{news} W_Q$, $K_{pirce} = Z_{price} W_K$, $V_{price} = Z_{pirc} W_V$
- $Q_{price} = Z_{price} W'_Q$, $K_{news} = Z_{news} W'_K$, $V_{news} = Z_{news} W'_V$

$$Z_F = \text{LayerNorm}(Z_{\text{price-news}} + Z_{\text{news-price}}) \tag{6}$$

These bidirectional representations are then normalized to stabilize training and consolidate into a unified token sequence $Z_F$. It is fed into the LLM, which leverages both price patterns and news semantics to output news-aware features $O^{\text{news}}$ for downstream state encoding in the Transformer-based feature extraction module.

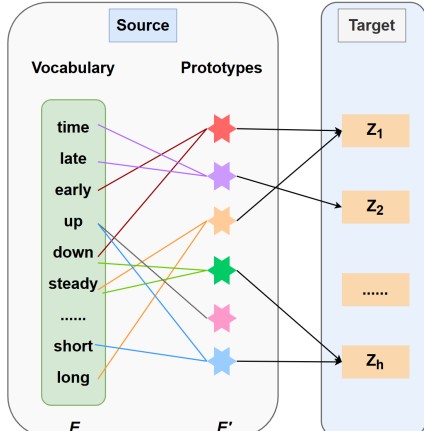

(a) Prompt example for financial tasks     (b) Reprogramming layer for price-LLM alignment

Figure 3: Key components of multimodal fusion.

## 2.2 TRANSFORMER-BASED FEATURE EXTRACTION

To address the limitation of traditional DRL in modeling multi-scale temporal dynamics and inter-stock dependencies, we design a Transformer-based feature extraction module to capture three key types of latent market signals: news trends, stock price fluctuations, and inter-stock correlation structures, specifically short/long-term news sentiment ($s^{news}_{short/long}$), short/long-term price trends ($s^{price}_{short/long}$), and stock correlations ($s^{relat}$). By leveraging the Transformer's self-attention mechanism, this module explicitly models the sequential dependencies in news and price data and the relational patterns between stocks, laying the foundation for robust multi-scale feature fusion and downstream policy learning.

## 2.3 MULTISCALE FUSION

Following transformer-based feature extraction, we first fuse the news textual spaces and price spaces across distinct temporal horizons, then integrate the price space with its associated correlation space, and finally consolidate these fused representations into a unified, comprehensive state space, resolving the misalignment between single-scale features and complex market dynamics:

$$s_t^{\text{news}} = \text{MH-Attn}(s_{\text{t,long}}^{\text{news}}, s_{\text{t,short}}^{\text{news}}, s_{\text{t,short}}^{\text{news}}) + s_{\text{t,long}}^{\text{news}}$$

$$s_t^{\text{price}} = \text{MH-Attn}(s_{\text{t,long}}^{\text{price}}, s_{\text{t,short}}^{\text{price}}, s_{\text{t,short}}^{\text{price}}) + s_{\text{t,long}}^{\text{price}} \quad (7)$$

$$s_t^{\text{hybrid}} = \text{MH-Attn}(s_t^{\text{price}}, s_t^{\text{relat}}, s_t^{\text{relat}}) + s_t^{\text{price}}$$

$$\mathcal{S}_t = \text{Concat}(s_t^{\text{hybrid}}, s_t^{\text{news}}, s_t^{\text{hold}}) \quad (8)$$

## 2.4 SENTIMENT-AWARE ADAPTIVE TRADING WITH SAC

We adopt the SAC algorithm for adaptive trading in $\mathcal{S}_t$ with a key innovation: In addition to estimating Q-values, the critic network propagates the gradients of state values back to the stock relation prediction module (as shown by the dashed line in 1. This enables joint training of both the predictive encoding and policy learning stages, where the critic's evaluation of state values facilitates further uncovering of correlations between trading assets from noisy, high-dimensional observations. The soft Q-function loss is:

$$\mathcal{J}(Q) = \mathbb{E}\left[\left(Q(s_t, a_t) - (R_t + \gamma(\hat{Q}_\tau - \lambda \log \pi))\right)^2 / 2\right] \quad (9)$$

## 3 EXPERIMENTS

### 3.1 DATASET DESCRIPTIONS

To avoid information leakage, all datasets strictly preserve temporal ordering, with training data drawn from earlier periods than their corresponding evaluation phases. The stock price data, is sourced from Yahoo Finance (refer to Supplementary Materials B.1). News data comes from the Hugging Face dataset (refer to Supplementary Materials B.1), covering news texts of Nasdaq 100 components. Due to significant missing news data, 10 stocks with complete news coverage between January 4, 2019 and December 30, 2022 were selected: ADBE, ADSK, ALGN, AMAT, AMD, AMGN, BIIB, CDNS, CHTR, and CMCSA.

All numerical features were normalized to [0, 1] using min-max scaling for model stability and feature consistency. Missing news texts were filled with "null" for subsequent identification and filtering; extra spaces, tabs, and line breaks were removed to enhance text standardization and processing efficiency.Additional data-related information is provided in the appendix.

### 3.2 IMPLEMENTATION DETAILS

Experiments were conducted on an Ubuntu 20.04 server equipped with an NVIDIA RTX 4090D GPU, with model development and training implemented using Python 3.8 and PyTorch 2.0.0.

For temporal multimodal LLM feature extraction, the framework processes 20-day stock price/volume time-series data combined with news headlines for multimodal fusion. It utilizes BERT/GPT-2 with a hidden dimension of 768, a maximum prompt length of 200, a dropout rate of 0.2 during fusion, and GeLU as the activation function. The training strategy employs a batch size of 8 over 50 epochs, with the Adam optimizer paired with OneCycleLR (maximum learning rate = 0.001) and early stopping (training terminates if no significant validation improvement is observed for 10 consecutive epochs). For task adaptation, the trained model is used in the inference phase to predict stock prices; for multi-stock trading tasks, it outputs 16-dimensional news feature vectors for the trading module.

The process employs both Transformer and SAC. The Transformer encoder uses 16-dimensional encoders/decoders for LLM feature input and 18 dimensions in correlation learning scenarios (10 stocks + 8 technical indicators), with modules trained for long- and short-term trend prediction and correlation feature inference. The SAC observation space includes covariance matrices, technical indicators (e.g., MACD, BOLL), and long- and short-term news/price indicators, with these features fused to inform trading decisions. Key parameters include a prediction sequence length of 20, a 63-day lookback period for covariance calculation, a SAC action space of 10 (corresponding to 10 stocks), and a 257-dimensional observation space after multi-head attention fusion.

### 3.3 COMPARATIVE METHODS AND EVALUATION METRICS

#### 3.3.1 MULTI-STOCK TRADING TASK

The buy-and-hold strategy based on the NASDAQ 100 Index serves as the benchmark. Additionally, 8 state-of-the-art stock trading approaches are included for comparison, comprising classic DRL algorithms, such as SAC, DDPG, TD3, A2C, and PPO, which are implemented using the FinRL framework (Li et al., 2021; Liu et al., 2021a), As comparative DRL baselines, the Ensemble Strategy (Yang et al., 2020), the Transformer-based TACR model (Lee & Moon, 2023), and the StockFormer model (Gao et al., 2023) are also included in this experiments.

When evaluating trading performance, five core indicators are adopted: Cumulative Return(CR), Annualized Return(AR), Sharpe Ratio(SR), Annualized Volatility (AV) and Maximum DrawDown (MDD). AV measures the degree of fluctuation in investment returns, and MDD reflects the maximum possible loss during the investment process. These indicators comprehensively measure the performance of the strategy.

### 3.3.2 STOCK PRICE PREDICTION TASK

To validate the performance gains from our proposed prompt engineering and reprogramming layer in stock price prediction, we select three representative baselines for comparison: Autoformer (time-series-focused), GPT-2, and BERT (LLM-focused), all lacking our multimodal alignment and task-specific optimizations. Autoformer (Wu et al., 2021) uses self-attention to analyze stock price time-series, capture trends/periodicity, and predict price trends. GPT-2 (Radford et al., 2019) is a large language model that mines market sentiment from text via text understanding. BERT (Devlin et al., 2019) adopts a bidirectional Transformer encoder to capture text context, learning text features via pre-training/finetuning for stock price prediction. In this experiment, they process the same news for prediction comparison. Prediction performance is evaluated using Mean Squared Error (MSE) and Mean Absolute Error (MAE) (see B.2 for details),which quantify prediction error for rigorous comparison.

### 3.4 EXPERIMENTAL RESULTS AND ANALYSIS

#### 3.4.1 MULTI-STOCK TRADING COMPARISON EXPERIMENT

To verify the effectiveness of the proposed temporal multimodal LLM stock trading strategy, experiments were conducted on 10 Nasdaq 100 component stocks with an initial capital of $100,000 (data in Table 1 and Figure 4 are based on this amount).

As shown in the multi-stock trading strategy comparison results in Table 1, during the period from December 2021 to December 2022, the temporal multimodal intelligent trading algorithms based on LLM (BERT/GPT-2) and Transformer demonstrated significant advantages, specifically as follows: **Performance of Return Generation**: Our proposed methods demonstrate standout performance

| Method | CR ↑ | AR ↑ | SR ↑ | AV ↓ | MDD ↓ |
|---|---|---|---|---|---|
| Buy-and-HoldFama & French (1993) | -0.315 | -0.297 | -0.944 | **0.320** | 0.355 |
| A2C(Liu et al., 2021a) | -0.349 | -0.329 | -1.088 | 0.321 | 0.421 |
| PPO(Liu et al., 2021a) | -0.336 | -0.316 | -0.632 | 0.447 | 0.445 |
| SAC(Liu et al., 2021a) | -0.505 | -0.480 | -1.610 | 0.365 | 0.550 |
| TD3(Liu et al., 2021a) | -0.438 | -0.415 | -1.040 | 0.428 | 0.513 |
| DDPG(Liu et al., 2021a) | -0.116 | -0.108 | -0.086 | 0.402 | 0.296 |
| Ensemble_Strategy(Yang et al., 2020) | -0.240 | -0.225 | -0.398 | 0.426 | 0.443 |
| TACR(Lee & Moon, 2023) | 0.026 | 0.024 | 0.261 | 0.456 | 0.326 |
| StockFormer(Gao et al., 2023) | 0.018 | 0.017 | 0.225 | 0.393 | 0.270 |
| **Ours(GPT2)** | 0.169 | 0.156 | 0.541 | 0.408 | **0.227** |
| **Ours(BERT)** | **0.191** | **0.164** | **0.544** | 0.440 | 0.244 |

Table 1: Performance comparisons of multi-stock trading strategies (Dec 2021 - Dec 2022)

across return generation, volatility adaptation, and risk-adjusted returns, critical for volatile markets, with Figure 4 validating these advantages dynamically: Among the proposed methods, our BERT-based algorithm achieved a cumulative return of 0.191 with an annualized return of 0.164; our GPT-2-based algorithm also obtained a positive return of 0.169 with an annualized return of 0.156. In contrast, the buy-and-hold strategy showed a significant negative return, with cumulative return at 0.315 and annualized return at -0.297. Traditional reinforcement learning methods generally failed to achieve positive returns. The Transformer-based StockFormer achieved a small positive return of 0.018, with annualized return at 0.017, but still lagged behind the proposed methods. This confirms our framework's superiority in generating returns by fusing news sentiment and price dynamics.

**Performance of Volatility Adaptation**: We evaluate volatility resilience via AV and MDD. Our BERT-based algorithm achieves an AV of 0.440 and a MDD of 0.244; Our GPT-2-based algorithm had an AV of 0.408 and a MDD of 0.227. This highlights our framework's ability to balance fluctuation control and loss mitigation in volatile markets.

**Performance of Risk-adjusted return indicators**: The SR (excess return per unit risk) further validates our advantage. Ours (BERT) and Ours (GPT-2) achieve 0.544 and 0.541, respectively, far exceeding all baselines. StockFormer's SR of 0.225 represents only 37% of the BERT-based

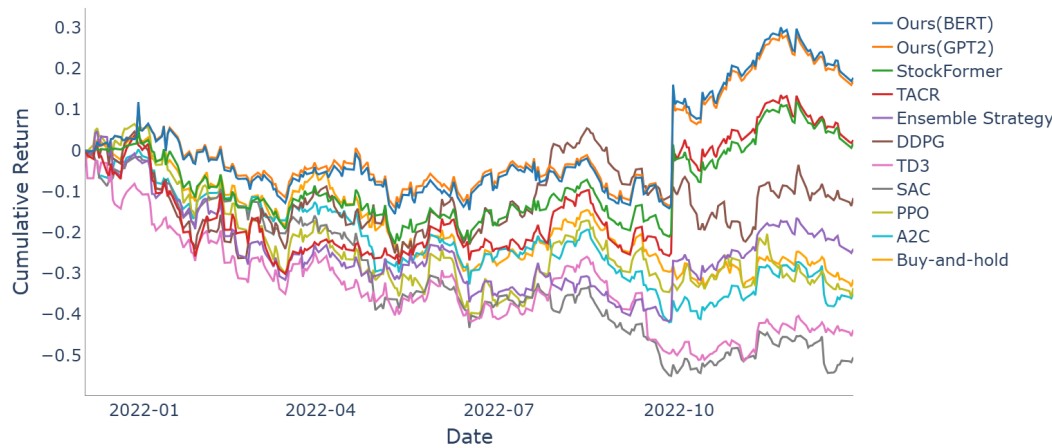

Figure 4: The cumulative returns of different trading strategies in the NASDAQ 100 market

method's result, highlighting the superior performance of the proposed approaches in terms of risk-adjusted returns.

Figure 4 shows the dynamic process of the daily cumulative returns of each algorithm during the backtest period, further validating the above conclusions. During periods of severe market fluctuations, the buy-and-hold strategy on the Nasdaq 100 index fell by 31.5%; DRL methods experienced systemic failure, with the SAC strategy suffering a loss of 50.5% during the same period, exposing the vulnerability of RL methods to black swan events. Although StockFormer achieved a positive return of 1.8%, its overall performance was still inferior to that of the proposed methods. Through effective extraction of text information and risk diversification, the proposed methods based on GPT-2 and BERT achieved high positive returns, capable of demonstrating significant volatility resistance, return sustainability, and strategy stability.

### 3.5 SINGLE-STOCK PREDICTION COMPARISON EXPERIMENT

The single-stock prediction comparison experiment is designed to evaluate the efficacy of the proposed temporal multimodal LLM module in predicting individual stock performance. The study focuses on five constituent stocks of the NASDAQ: **ADSK**, **ALGN**, **AMD**, **CHTR**, and **CMCSA**.

The comparative analysis incorporates the following benchmark methods: the time-series prediction model Autoformer, as well as the pre-trained models BERT and GPT-2. As illustrated in Table

Table 2: The comparison of single-stock prediction experimental results

| Method | ADSK | | ALGN | | AMD | | CHTR | | CMCSA | |
|---|---|---|---|---|---|---|---|---|---|---|
| | MSE↓ | MAE↓ | MSE↓ | MAE↓ | MSE↓ | MAE↓ | MSE↓ | MAE↓ | MSE↓ | MAE↓ |
| Autoformer(Wu et al., 2021) | 0.46 | 0.53 | 1.69 | 1.04 | 9.66 | 3.02 | 5.95 | 2.26 | 2.43 | 1.26 |
| GPT-2(Radford et al., 2019) | 0.64 | 0.74 | 0.67 | 0.76 | 0.77 | 0.71 | 3.07 | 1.61 | 1.03 | 0.82 |
| BERT(Devlin et al., 2019) | 0.31 | 0.49 | 0.71 | 0.78 | 1.22 | 0.97 | 0.69 | 0.70 | 1.22 | 0.91 |
| **Ours(GPT-2)** | 0.29 | 0.47 | **0.39** | **0.57** | **0.33** | **0.48** | 2.49 | 1.42 | **0.75** | **0.73** |
| **Ours(BERT)** | **0.26** | **0.45** | 0.58 | 0.71 | 0.79 | 0.72 | **0.54** | **0.62** | 1.12 | 0.87 |

4, Ours methods (GPT-2 and BERT) exhibit superior performance in forecasting multiple stocks. Specifically, for ADSK and CHTR, the Ours (BERT) model achieves lower prediction errors than alternative methods, thereby demonstrating enhanced accuracy; for ALGN, AMD, and CMCSA, the Ours (GPT-2) model delivers more precise predictions.

### 3.6 ABLATION EXPERIMENTS

This series of ablation experiments selects BERT as the base model and is conducted on multi-stock trading tasks and stock prediction tasks. Specifically, it focuses on examining the effectiveness of each module and the feature fusion mechanisms therein in trading tasks, as well as the influence of the sophistication of LLMs prompt engineering and reprogramming layer on prediction results.

#### 3.6.1 ABLATION STUDY ON TRADING MODULES

As demonstrated in Table 3, we assess the individual and combined impacts of four core modules: News prediction module, Price prediction module, Correlation inference module and the critical role of the attention-based cross-modal fusion mechanism.

Table 3: The ablation study of core modules

| News Prediction Module | Price Prediction Module | Correlation Inference Module | Feature Fusion Mechanism | CR | SR |
|---|---|---|---|---|---|
| | ✓ | ✓ | ✓ | 0.018 | 0.225 |
| ✓ | | ✓ | ✓ | 0.147 | 0.483 |
| ✓ | ✓ | | ✓ | 0.185 | 0.582 |
| ✓ | ✓ | ✓ | | 0.167 | 0.530 |
| ✓ | ✓ | ✓ | ✓ | **0.191** | **0.608** |

The complete model exhibits superior performance: it achieves 3.2% higher cumulative returns than the best sub-module combination, which integrates news and price data; a 4.5% improvement in the Sharpe Ratio; and a 10.6-fold return enhancement over the baseline that combines price and correlation data, with returns increasing from 0.018 to 0.191.

These results validate that the news prediction module effectively synthesizes multi-source information to enhance market trend analysis and trading decisions. The ablation experiment also reveals a significant performance degradation when the attention mechanism is removed: cumulative returns decrease by 12.6%, falling from 0.191 to 0.167, while the Sharpe Ratio drops by 12.8%, declining from 0.608 to 0.530.

Additional ablation study and supplementary statistical tests are discussed in Supplementary Materials (refer to 7). We firstly evaluate the contribution of each component to prediction performance(e.g., LLMs and the reprogramming layer); We also provide the details of the statistically significant difference between the proposed approach and random strategies.

## 4 CONCLUSION

This paper introduces a multimodal DRL framework that integrates LLMs and Transformers to address key limitations in existing approaches: the underutilization of multimodal data, misalignment between textual information and price movements, and insufficient modeling of volatility and multi-scale market dynamics. A multimodal LLM module is employed to fuse news and price data through prompt engineering and reprogramming layers. Additionally, a Transformer-based model is developed to extract and combine features, which are jointly trained with a SAC algorithm to strengthen the synergy between feature representation and policy learning.

Experimental results show that the proposed method outperforms benchmark models across key performance metrics, including the SR and MDD, in both multi-stock trading and single-stock prediction tasks. Ablation studies confirm the contribution of each component within the trading and prediction modules, as well as the performance improvement attributable to the reprogramming layer. Statistical tests further demonstrate a significant difference between the proposed strategy and random strategies, underscoring the robustness of our approach.

In summary, this framework advances the integration of multimodal data and strategy optimization, offering valuable insights for stock trading research and practical support for investors in volatile markets. A promising future direction involves exploring multi-agent collaboration and specialization to improve operational efficiency while mitigating the effects of LLM hallucinations.

## 5 ETHICS STATEMENT

This study develops a multimodal deep reinforcement learning framework for the financial domain, with adherence to ethical principles detailed as follows:

**Data Ethics** All datasets employed are publicly accessible resources. Their usage strictly complies with the terms of service set forth by the data providers. No sensitive information is included in the datasets; missing news data is filled with "null" solely to ensure the smooth progression of experiments, which is in line with relevant data privacy guidelines.

**Research Integrity** To avoid information leakage, the dataset is split into training and test sets, each belonging to distinct time periods. The model architecture, hyperparameters, and computing infrastructure have all been fully disclosed. Experimental results are presented objectively in this study, with clear distinction from the limitations of the framework, and no exaggerated claims are made.

**Market Impact** This framework is intended exclusively for academic research and shall not be used for live commercial trading. Guided by the core principle of responsible innovation, this study aims to advance knowledge at the intersection of artificial intelligence and finance while minimizing potential risks to markets and society.

## 6 REPRODUCIBILITY STATEMENT

To ensure the reproducibility of this study's findings, all critical resources and details have been systematically organized and documented in the main text, appendix, and supplementary materials. For the framework proposed in this study, the complete implementation code, pre-trained models, and evaluation scripts will be made publicly available upon the paper's acceptance. This codebase includes full training and testing scripts for multimodal LLM analysis, Transformer inference, and SAC-based trading implementation, as well as all experimental configurations utilized in this research—facilitating researchers' convenient access to replicate the model training and prediction processes.

Regarding the datasets used in the experiments, the core dataset division principles and key statistical characteristics are outlined in the main text and supplementary materials. The supplementary materials further provide detailed descriptions of data sources and links to open-source dataset repositories, enabling the replication of data acquisition and preparation workflows. For model parameters and experimental settings, Section 3.2 of the main text specifies the key hyperparameters of the experiments, while the appendix supplements additional critical configuration details—eliminating ambiguities in experimental setup.

Additionally, the ablation studies and statistical tests conducted to validate the effectiveness of model components and the significance of results are thoroughly documented in the experimental section and appendix. These materials include complete raw result tables and statistical test metrics, allowing readers to verify the robustness of this study's conclusions. By cross-referencing the aforementioned resources, researchers can fully replicate the entire experimental workflow—from data preparation and model training to result validation—ensuring that the performance metrics and core findings of this study are independently verifiable.

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

## A    USE OF LLMs

LLMs were employed in a limited capacity during the writing of this manuscript, solely to assist with textual refinement for logical coherence, terminological standardization, and overall clarity. All core intellectual content, including experiments, model designs, and results, was produced by the authors. This use aligns with academic integrity principles, and the authors assume complete responsibility for the work's authenticity.

## B    SUPPLEMENTARY MATERIALS

### B.1    CODE, DATASETS AND IMPLEMENTATIONS

**Code** To ensure the reproducibility of our results, the code supporting this study will be made publicly available on GitHub shortly after the paper is accepted.

**Dataset** This paper utilizes multiple publicly available datasets. Below is an explanation of the usage and sources of each dataset. We evaluate our framework on stocks from the NASDAQ-100 index and a publicly available stock news dataset:

**STOCKS**: ADBE, ADSK, ALGN3, AMAT, AMD, AMGN, BIIB, CDNS, CHTR, CMCSA **nasdaq_news**: 20,580 images of 120 dog breeds

Both datasets are publicly accessible via their official sources. The stock price data can be accessed on Yahoo Finance.

https://finance.yahoo.com

while the news data comes from an open-source dataset

https://huggingface.co/datasets/benstaf/nasdaq_news

**Data division for training and testing**

Table 4: Dataset Division of the Time-Series LLM Prediction Module

| Dataset Type | Start Date | End Date | Trading Days |
|---|---|---|---|
| Training Set | 2019-01-04 | 2021-12-30 | 735 |
| Validation Set | 2020-12-30 | 2022-01-03 | 236 |
| Testing Set | 2021-12-30 | 2022-12-30 | 234 |

For pre-training the temporal multimodal LLM prediction module, the dataset was split into training, validation, and test sets to enable the model to learn data patterns and verify generalization. As shown in Table 4, data was split strictly by time to avoid future information leakage: 735 trading days for the training set (covering various market cycles), 236 days for the validation set, and 234 days for the test set.

Given the limited sample size, where all stock market-related information is critical for model training, the pre-trained data is split into training and test sets to maximize utilization of such information. As detailed in Table 5. We also attempted to conduct experimental research on Chinese stock market data; however, there is currently a lack of standardized Chinese news text data for the stock market. This presents a promising avenue for future exploration.

Table 5: Dataset Division of the Trading Module

| Dataset Type | Start Date | End Date | Trading Days |
|---|---|---|---|
| Training Set | 2019-05-03 | 2021-11-04 | 643 |
| Testing Set | 2021-11-04 | 2022-12-30 | 290 |

**Implementation**

Our implementation is built on the FinRL library, which provides a comprehensive framework for reinforcement learning strategies in stock trading

https://github.com/AI4Finance-Foundation/FinRL

including data preprocessing, pre-trained models, and trading strategy optimization.

Our multimodal fusion model based on LLM, which is built upon the implementation of the paper - TimeLLM(Jin et al., 2023)

https://github.com/KimMeen/Time-LLM

We primarily made adjustments for stock price sequences. Specifically, we map the massive token vocabulary of the native large model to a 1000 - dimensional space using input financial news data, thus focusing on the financial domain and achieving alignment with price features through reprogramming.

Our codebase extends the reinforcement learning state by analyzing news data with multimodal large models and predicting stock prices using Transformer, thereby achieving trading strategy optimization and reinforcement learning performance improvement.

### B.2 IMPLEMENTATION DETAILS AND HYPERPARAMETERS

Table 6 summarizes the key implementation details and hyperparameters employed in all experiments. Further details and configuration files are available in the supplementary repository.

Table 6: Summary of key implementation details and hyper-parameters for the stock prediction and trading model.

| Component | Setting / Value |
|---|---|
| Framework | PyTorch |
| GPU | NVIDIA RTX 4090D |
| Language Models | BERT, GPT-2 |
| llm_dim | 768 |
| Prompt Max Length | 200 |
| dropout (for fusion) | 0.2 |
| Activation Function | GeLU |
| Batch Size (Training) | 8 |
| Training Epochs | 50 |
| Optimizer | Adam |
| Learning Rate Scheduler | OneCycleLR, max_learning_rate=0.001 |
| Early Stopping Criterion | 10 consecutive epochs without performance gain on validation set |
| **Transformer Prediction Encoding (LLM features)** | |
| Encoder/Decoder Input | dim=16 |
| **Transformer Prediction Encoding (correlation feature)** | |
| Number of Stocks | 10 |
| Technical Indicator | dim=8 |
| Encoder/Decoder Input | dim=18 |
| **Transformer Prediction Module** | |
| Prediction Sequence | Length=20 |
| look_back | 63 |
| **SAC Module** | |
| Action Space | 10 |
| Observation Space | |
| - Covariance matrix | dim=1 |
| - Technical indicators | dim=8 |
| - Short-long term news & price prediction encoding | dim=4$\times$128 |
| - Position state | dim=1 |
| Comprehensive Feature Space (after fusion) | dim=257 |

The formulas used to evaluate the prediction performance are MSE and MAE, defined as follows:

$$MSE = \frac{1}{N} \sum_{i=1}^{N} (y_i - \hat{y}_i)^2 \tag{10}$$

$$MAE = \frac{1}{N} \sum_{i=1}^{N} |y_i - \hat{y}_i| \tag{11}$$

## C   ABLATION STUDY

This section is a supplement to the ablation study. Using CHTR stock data, Table 7 evaluates the impact of individual prompt components and the reprogramming layer in the temporal multimodal LLM:

Table 7: The ablation study of multimodel LLM components

| Method | MSE ↓ | MAE ↓ |
|---|---|---|
| Without Dataset Description | 1.103 | 0.884 |
| Without Task Description | 1.087 | 0.870 |
| Without News Information Description | 1.053 | 0.856 |
| Without Reprogramming Layer | 1.100 | 0.875 |
| **Ours** | **0.191** | **0.608** |

Key findings include the following: Removing the dataset description leads to a 4.77-fold increase in MSE, which highlights its role in initializing domain knowledge; removing the task description results in a 4.69-fold deterioration in MSE, emphasizing the necessity of explicit objective alignment; excluding the news context causes a 40.8% increase in MAE, confirming its value in capturing trends; and ablating the reprogramming layer brings about a 4.75-fold deterioration in MSE, with the metric rising from 0.191 to 1.100, thus proving its effectiveness in noise suppression.

## D   STATISTIC TEST

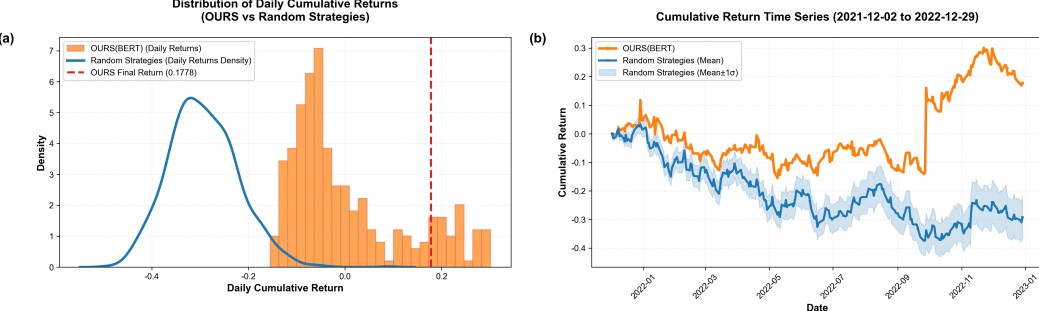

Figure 5: Daily Cumulative Returns Distribution of OURS(BERT) vs Random Strategies and Cumulative Return Time Series from 2021-12-02 to 2022-12-29 (OURS(BERT) vs Random Strategies)

We also performed a significance test to statistically compare the cumulative return sequence from the experiment with the return sequence generated by random trading. This comparison was conducted to verify whether the distribution of experimental outcomes significantly differs from that of a random strategy.

As shown in Figure 5 a, it presents the probability density distributions of cumulative returns for the random strategy and the framework in this study. With an average cumulative return rate of 0.1778, the performance of this study is significantly superior to the random strategy, whose average cumulative return rate stands at -0.2929. Considering that the $p$ value of the Kolmogorov - Smirnov test

is 0.0, the following null hypothesis is rejected at a significance level of $\alpha = 0.01$: "The distribution of our framework is the same as that of random trading strategies".

Figure 5 b offers a straightforward comparison of the cumulative return rate of this study versus the average return rate of the random strategy. Beyond the initial short period where their performance is nearly indistinguishable, our experimental framework maintains stable returns afterward and even demonstrates considerable profitability in bear market scenarios.

## E  ALGORITHMIC DETAILS

Algorithm 1, and Algorithm 2 provide pseudocodes for state space expansion through feature fusion, respectively, and stock trend generation using multimodal large models .

---

**Algorithm 1** Multimodal DRL Trading Algorithm

---

**Require:** Observation Space: $O^{\text{news}}, O^{\text{price}}, O^{\text{rela}}, s_{\text{hold}}$, Market Volatility $\sigma_t$
**Ensure:** State Space $S$ with uncertainty metric $\epsilon$
 1: **I. LLM-Guided Feature Preprocessing**
 2: $O_{\text{short}}^{\text{price}} \leftarrow \text{ExtractWindow}(O^{\text{price}}, 60)$
 3: $O_{\text{long}}^{\text{price}} \leftarrow \text{ExtractWindow}(O^{\text{price}}, 60)$
 4: **II. Adaptive Prediction Module**
 5: *Price Encoder:*
 6: $enc_{\text{short}} \leftarrow \text{TransformerEncoder}(O_{\text{short}}^{\text{price}}, \text{attn\_span} = 1)$
 7: $enc_{\text{long}} \leftarrow \text{TransformerEncoder}(O_{\text{long}}^{\text{price}}, \text{attn\_span} = 5)$
 8: *News Encoder:*
 9: $enc_{\text{newsshort}} \leftarrow \text{TransformerEncoder}(O_{\text{short}}^{\text{news}}, \text{attn\_span} = 1)$
10: $enc_{\text{newslong}} \leftarrow \text{TransformerEncoder}(O_{\text{long}}^{\text{news}}, \text{attn\_span} = 5)$
11: *Correlation Encoder:*
12: $s_{\text{relat}} \leftarrow \text{SparseTransformer}(O^{\text{rela}}, k = 5)$
13: **III. Dynamic Feature Fusion**
14: $s_{\text{price}} \leftarrow \text{MHA}(enc_{\text{long}}, enc_{\text{short}}, enc_{\text{short}}) + enc_{\text{long}}$
15: $s_{\text{news}} \leftarrow \text{MHA}(enc_{\text{newslong}}, enc_{\text{newsshort}}, enc_{\text{newsshort}}) + enc_{\text{newslong}}$
16: $s_{\text{hybrid}} \leftarrow \text{MHA}(s_{\text{price}}, s_{\text{relat}}, s_{\text{relat}}) + s_{\text{price}}$
17: **IV. State Space Construction**
18: $\boldsymbol{S} = \text{Concat}(s_{\text{hybrid}}, \boldsymbol{s}_{\text{hold}}, s_{\text{news}}),$
19: return $\boldsymbol{S}$

---

---

**Algorithm 2** Stock Trend Prediction via Multimodal Fusion Based on LLMs

---

**Require:** Raw time-series $D = \{o_t, c_t, h_t, l_t, v_t\}$ ($t = 1, \ldots, T$), Text news, LLMs BERT/GPT-2, LLM word embedding matrix $E \in \mathbb{R}^{V \times D}$
**Ensure:** Multimodal latent representation $Z_{\text{fusion}}$ (for downstream Transformer/SAC)
1: **1. Price Preprocessing and Multiscale Embedding**
2: Normalize $D$ to $\hat{D}$ using min-max scaling
3: Define short/long windows: $L_{\text{short}} = 1$, $L_{\text{long}} = 5$
4: $N_{\text{short}} = \lfloor T/L_{\text{short}} \rfloor$, $N_{\text{long}} = \lfloor T/L_{\text{long}} \rfloor$
5: Split $\hat{D}$ into short/long patches:
6: $P_{\text{short}} = \{p_1^{\text{s}}, \ldots, p_{N_{\text{short}}}^{\text{s}}\}$, $p_i^{\text{s}} = \{\hat{D}_{(i-1)L_{\text{short}}+1}, \ldots, \hat{D}_{iL_{\text{short}}}\}$
7: $P_{\text{long}} = \{p_1^{\text{l}}, \ldots, p_{N_{\text{long}}}^{\text{l}}\}$, $p_i^{\text{l}} = \{\hat{D}_{(i-1)L_{\text{long}}+1}, \ldots, \hat{D}_{iL_{\text{long}}}\}$
8: Encode patches with linear projection (preserve temporal structure, ):
9: **for** each $p_i^{\text{s}} \in P_{\text{short}}$ **do**
10:     $e_i^{\text{s}} = \text{LinearProj}(p_i^{\text{s}}) \in \mathbb{R}^D$
11: **end for**
12: **for** each $p_i^{\text{l}} \in P_{\text{long}}$ **do**
13:     $e_i^{\text{l}} = \text{LinearProj}(p_i^{\text{l}}) \in \mathbb{R}^D$
14: **end for**
15: $E_{\text{price}} = \text{Concat}(E_{\text{short}}, E_{\text{long}})$ where $E_{\text{short}} = \{e_i^{\text{s}}\}$, $E_{\text{long}} = \{e_i^{\text{l}}\}$
16: **2. News Encoding with Prompt Engineering**
17: Tokenize news: tokens = Tokenizer(news)
18: Encode tokens: $text_{\text{emb}} = \{W(t) \mid t \in \text{tokens}\} \in \mathbb{R}^{n \times D}$
19: Construct task-specific prompt ():
20: prompt = *"Predict stock closing price. Use: "* + history prices + *" and news: "* + news content

21: Encode news with LLM and prompt: $Z_{\text{news}} = \text{LLM}(\text{Concat}(\text{prompt\_emb}, text_{\text{emb}})) \in \mathbb{R}^{n \times D}$
22: **3. Price Reprogramming to LLM Space**
23: $E_{\text{ref}} \leftarrow E$
24: **for** Each attention head $h$ **do**
25:     Query: $Q_h = X_{\text{price}} W_h^Q$
26:     Key: $K_h = E_{\text{ref}} W_h^K$
27:     Value: $V_h = E_{\text{ref}} W_h^V$
28: **end for**
29: $Z_h = \text{SoftMax}(Q_h K_h^T / \sqrt{D}) V_h$
30: $Z_{\text{price}}^{\text{concat}} = \text{Concat}(Z_h) W^O$
31: $Z_{\text{price}} = \text{LayerNorm}(Z_{\text{price}}^{\text{concat}})$
32: **4. Cross-Modal Fusion**
33: $Q_{\text{news}} = Z_{\text{news}} W_Q$, $K_{\text{price}} = Z_{\text{price}} W_K$, $V_{\text{price}} = Z_{\text{price}} W_V$
34: $Z_{\text{price-news}} = \text{SoftMax}\left(\frac{Q_{\text{news}} K_{\text{price}}^T}{\sqrt{D}}\right) V_{\text{price}}$
35: $Q_{\text{price}} = Z_{\text{price}} W_Q'$, $K_{\text{news}} = Z_{\text{news}} W_K'$, $V_{\text{news}} = Z_{\text{news}} W_V'$
36: $Z_{\text{news-price}} = \text{SoftMax}\left(\frac{Q_{\text{price}} K_{\text{news}}^T}{\sqrt{D}}\right) V_{\text{news}}$
37: $Z_{\text{fusion}} = \text{LayerNorm}(Z_{\text{price-news}} + Z_{\text{news-price}})$
38: return $Z_{\text{fusion}}$

---

