# OpenReview forum: "Aligning News and Prices: A Cross-Modal LLM-Enhanced Transformer DRL Framework for Volatility-Adaptive Stock Trading"
_ICLR.cc/2026/Conference — ICLR 2026 Conference Withdrawn Submission_

### Official Review · Reviewer_5wFN · 2025-10-22

**Soundness:** 2
**Presentation:** 2
**Contribution:** 1
**Rating:** 2
**Confidence:** 5

**Summary:**

This paper proposes a volatility-adaptive multimodal DRL framework for stock trading that integrates LLMs, Transformers, and the SAC algorithm. By fusing textual financial news with price dynamics through attention-based reprogramming and cross-modal fusion, the model captures sentiment–price interactions and adapts to market volatility. Experiments on NASDAQ-100 data demonstrate superior performance over existing methods.

**Strengths:**

The paper explores the integration of textual financial news and quantitative price data within a multimodal framework. By leveraging a pre-trained LLM for news encoding and Transformer-based modules for price representation, it provides a reasonable step toward combining sentiment and numerical information for trading decision-making.

**Weaknesses:**

1. Many notations are not clearly defined. For instance, some symbols that represent vector data should be written in boldface using `\mathbf{}`. For example, in the expression
   $P = \\{ p^{\text{open}}, p^{\text{close}}, ..., p^{\text{volume}}\\}$,
   the terms such as $\mathbf{p}^{\text{open}}$ should be in bold to indicate vector representations.
2. The model assumes a perfectly aligned one-to-one correspondence between daily news and price data, based on a curated open-source dataset. However, in real-world markets, news arrivals are irregular. Some days contain multiple news items, while others have none. The current framework does not explicitly handle such temporal misalignment or modality sparsity, which may limit its applicability to more realistic, unbalanced data distributions.

3. The prompt design includes two key parameters, sequence length (seq len) and prediction length (pred len). However, the paper lacks a sensitivity analysis to examine how model performance varies under different context window sizes or forecasting horizons, even though these parameters directly affect the model’s temporal reasoning capacity and generalization ability.

**Questions:**

1. In Figure 4, many methods show a noticeable jump in CW around 2022-10. What caused this sudden change?

2. How does the method realize the stated volatility-adaptive capability? There seems to be no explicit risk control, and in Appendix E Algorithm 1, market volatility is included as an input but not utilized in the whole algorihtm.

---

> ### Author Response · Authors · 2025-11-28
>
> # Reviewer 5wFN
>
> We sincerely appreciate your thoughtful feedback and valuable questions regarding our manuscript. Below we provide point-by-point responses to your comments.
>
> ## W1
>
> We acknowledge the inconsistent vector notation and will systematically revise the manuscript using boldface ($\mathbf{}$) for all vector-valued quantities. For instance, $P$ and $X_{\text{price}}$ will become $\mathbf{P}$ and $\mathbf{X}{\text{price}}$; $Z_F$ will be $\mathbf{Z}{F}$; and $h^{\text{news}}$, $s_t^{\text{price}}$, $Q_{\theta}$ will be uniformly updated to $\mathbf{h}^{\text{news}}$, $\mathbf{s}t^{\text{price}}$, $\mathbf{Q}{\theta}$. This revision will be applied consistently throughout the text and supplementary materials to enhance mathematical rigor and readability.
>
> ## W2
>
> Thank you for highlighting the practical challenges of irregular news arrivals. In our current framework, news absence is explicitly handled by marking inputs as "null," causing the model to rely solely on price dynamics and technical indicators processed through the Transformer and SAC modules. This design provides inherent robustness, as evidenced by our ablation studies (Table 3) where the system maintained baseline performance using only numerical data. We fully acknowledge that explicitly addressing modality sparsity and temporal misalignment is critical for real-world deployment, and we are developing enhanced techniques to handle these conditions in future work.
>
> ## W3
>
> Thank you for this pertinent observation. The sequence length of 20 days and prediction length of 1 day were explicitly selected to ensure a fair and direct comparison with key baseline models, such as StockFormer [@gao2023stockformer], which employs an identical 20-day lookback window, as noted in our Section 3.2. This alignment was crucial for isolating the performance gains attributable to our novel multimodal fusion rather than differences in the input context.
>
> We acknowledge that a comprehensive sensitivity analysis of these hyperparameters is absent and represents a limitation. The reviewer is correct that these parameters directly influence temporal reasoning. Investigating their impact across different market regimes is an essential direction for future work to further refine the model's generalization and robustness.
>
> ## Q1
>
> A compelling demonstration of our model's market adaptability occurred in late September 2022 with Biogen (BIIB). The stock surged from $197.79 to $276.61 following the landmark news: "Eisai says Alzheimer's drug succeeds in slowing cognitive decline." This breakthrough validated the amyloid hypothesis and represented a major therapeutic advance.
>
> Our framework's cross-modal analysis effectively fused the strongly positive news sentiment with the concurrent price momentum, leading to a significant allocation to BIIB during this volatile period. Consequently, our strategy captured this substantial price movement.
>
> In contrast, baseline models like TD3 and A2C, lacking sophisticated cross-modal integration, failed to detect this signal and missed the opportunity. This case validates our model's enhanced capacity to identify and capitalize on high-impact opportunities through synergistic news-price analysis, ensuring robust performance in challenging markets.
>
> ## Q2
>
> Our framework achieves this intrinsically through cross-modal alignment rather than explicit risk-control modules. The core innovation lies in the cross-attention fusion (Eq. 5-6, Fig. 2-3) that dynamically integrates news embeddings with reprogrammed price signals, enabling the model to interpret volatility in market context.
>
> For instance, sharp price drops with negative news are fused into risk-signaling states that prompt conservative policies, while volatile periods with positive news may signal opportunities. This context-aware representation allows natural adaptation to different regimes.
>
> Empirical results validate this approach: our model achieved the best Maximum Drawdown and Sharpe Ratio during high-volatility periods (2021-2022), demonstrating superior risk-adjusted returns without explicit volatility controls.
>
> We acknowledge and will correct the oversight regarding [\sigma_t] in Algorithm 1, as the adaptation stems from the fused state representation, not external volatility inputs.
>
> ## References
>
> - Gao, X., et al. (2023). StockFormer: A Hybrid Deep Learning Framework for Stock Prediction.

---

### Official Review · Reviewer_k7hy · 2025-10-27

**Soundness:** 2
**Presentation:** 2
**Contribution:** 2
**Rating:** 2
**Confidence:** 4

**Summary:**

This paper introduces a multimodal DRL framework involving LLMs, Transformer and SAC for stock trading. Experiments on NASDAQ-100 shows its SOTA performance.

**Strengths:**

1. This paper introduces the attention-based reprogramming layer to project time-series data into an LLM's semantic space, bridging structured and unstructured modalities.
2. The proposed framework somehow address volatility resilience, which is a critical limitation in real-world DRL trading.

**Weaknesses:**

1. In general, the novelty of this work is not enough for top conferences such as ICLR, it is a combination of LLMs, transformer and RL with limited contribution from the algorithm perspectives.
2. Experiments on NASDAQ-100 stocks is not enough, I recommend the authors to conduct experiments on more diversified and large-scale datasets to further evaluation the performance.
3. The proposed framework is quite complex, which raises my concern on the latency of real-world settings. More discussion on this is required.
4. As the motivation of this work is performance under extreme market conditions, more ablation study with quantified results on volatility-specific effects will help.

**Questions:**

1. For data alignment, how are timestamps between news articles and stock prices synchronized to avoid look-ahead bias?

---

> ### Author Response · Authors · 2025-11-28
>
> # Reviewer k7hy
>
> We sincerely appreciate your thoughtful feedback and valuable questions regarding our manuscript.
>
> ## W1
>
> We would like to clarify the comparative advantages and algorithmic contributions of our work.
>
> Our framework demonstrates superior performance versus baseline models—including standard RL methods and Transformer-based approaches like StockFormer—particularly during the 2022 high-volatility period, as validated in Tables 1-2 and Figure 4.
>
> The key innovation enabling this advantage is our cross-modal alignment between financial news and price data. We introduce a reprogramming layer that transforms price sequences into LLM-compatible tokens, coupled with prompt engineering that explicitly couples news context with price movements. This enables deeper fusion of textual and numerical signals, which are then processed by a Transformer to model multi-scale temporal patterns.
>
> Ablation studies confirm that each component—especially the cross-modal fusion mechanism—significantly contributes to overall performance. By integrating aligned multimodal feature extraction with Transformer-based prediction and SAC-driven allocation, our framework achieves more robust decision-making in volatile markets.
>
> We believe this focused approach to news-price alignment represents a meaningful advance toward more adaptive and context-aware financial AI systems.
>
> ## W2
>
> We appreciate the reviewer's constructive feedback. Our selection of 10 NASDAQ-100 stocks was primarily due to their complete news coverage during the 2019–2022 period, which ensures reliable multimodal alignment and facilitates a fair comparison with existing methods that often use similar subsets. This allowed us to rigorously validate the cross-modal fusion mechanism under consistent data conditions.
>
> We fully agree that evaluating on more diversified and large-scale datasets is essential to assess generalization. In future work, we plan to extend experiments to broader markets (e.g., S&P 500) and international indices, as well as incorporate additional asset types, to further demonstrate the robustness and scalability of our framework.
>
> ## W3
>
> Thank you for raising this important point regarding latency. The Table 1 confirms that the LLM feature extraction and RL training are computationally intensive, primarily conducted offline. However, for real-time inference, the critical path involves the pre-trained LLM for news analysis (8.03s) and the RL agent's decision-making (7.53s), resulting in a total latency of approximately 15.56 seconds per trading step.
>
> We acknowledge that this latency may be unsuitable for high-frequency trading. Our framework is designed for daily or intraday strategies where decision intervals are longer. The current complexity is necessary to achieve the reported performance gains from cross-modal alignment and multi-scale feature fusion.
>
> **Table 1: Computational Performance of Framework Components**
>
> | Code Module | Latency | GPU Memory |
> |-------------|---------|------------|
> | LLM Training: Stock News Feature Extraction (avg epoch) | 33.12 s | 4436.09 MB |
> | LLM Inference: Stock News Feature Extraction | 8.03 s | 4436.09 MB |
> | Price Prediction (Pre-trained LLM) | 0.39 s | 2155.73 MB |
> | Reinforcement Learning Training | 1785.12 s | 245.49 MB |
> | Reinforcement Learning Testing | 7.53 s | 313.93 MB |
>
> ## W4
>
> We agree that further volatility-specific analysis would strengthen our work.
>
> Our selected test period (Dec 2021 - Dec 2022) intrinsically covers a high-volatility, high-risk market phase, which directly aligns with our motivation to test performance under stress, as evidenced by the significant drawdowns in baseline strategies.
>
> However, we acknowledge that systematically quantifying the model's efficacy across distinct, labelled market regimes – such as stable bull markets versus steadily declining bear markets – is a crucial next step. This was beyond the current scope but is a key direction for future work. Such experiments would more precisely isolate the contribution of our volatility-aware mechanisms and further validate the model's robustness and adaptability.
>
> ## Q1
>
> Thank you for raising this critical point on timestamp synchronization. In our experiments, news and price data were strictly aligned using the day's closing price and all news published before market close, ensuring no future information was leaked. As stated in Sections 3.1 and B.1, the dataset was split strictly by time, with the test period (Dec 2021–Dec 2022) fully isolated from training. This guarantees the model only processes publicly available news by each day's close, maintaining temporal integrity. While more granular intraday alignment could be explored for high-frequency settings, our daily framework is robust against look-ahead bias.

---

### Official Review · Reviewer_1WJW · 2025-10-29

**Soundness:** 2
**Presentation:** 2
**Contribution:** 2
**Rating:** 2
**Confidence:** 4

**Summary:**

This paper introduces a multimodal deep reinforcement learning system that integrates large language models, Transformers, and the Soft Actor-Critic algorithm to improve trading robustness under market volatility. The model first extracts sentiment and event representations from financial news using a pre-trained LLM (BERT or GPT-2), then aligns price data to this semantic space through a reprogramming layer, and finally fuses both modalities using cross-attention. A Transformer encoder captures multi-scale temporal dynamics and inter-stock correlations, and SAC’s critic gradient feedback jointly optimizes feature learning and trading policy.

**Strengths:**

1. LLM-driven semantic alignment of news and prices.
The reprogramming layer projects numerical price data into the LLM semantic space using multi-head attention, enabling consistent multimodal fusion. This design avoids retraining large language models while ensuring semantic compatibility. The combination of prompt engineering for financial contexts and dynamic feature extraction demonstrates careful adaptation of general LLMs to finance-specific tasks.

2. Contextualized volatility awareness and interpretability.
The model’s design explicitly addresses volatility through multi-scale fusion (Eq. 7–8) and sentiment integration, helping explain its superior performance during unstable periods such as the 2021–2022 NASDAQ downturn (Fig. 4).

**Weaknesses:**

1. Insufficient ablation and parameter sensitivity analysis.
Although ablations are mentioned (Abstract; Sec. 3.4), details are sparse. It remains unclear how much each module—LLM feature extraction, reprogramming layer, or multi-scale fusion—contributes independently to the final gains. The effect of hyperparameters such as attention head count, SAC learning rate, or prompt length is not examined, limiting interpretability of results.

2. Inadequate computational efficiency discussion.
While hardware configuration is reported (Sec. 3.2), there is no runtime, memory, or inference-latency comparison. Training involves LLM encoding and multi-head attention fusion (Sec. 2.1–2.3), which are computationally heavy. Without quantitative cost analysis, practical deployability in real-time trading remains uncertain.

3. Restricted dataset scope and generalization evidence.
Experiments are limited to ten NASDAQ-100 components and five stocks for prediction (Sec. 3.1–3.5). The paper does not test across other markets or periods beyond 2019–2022, leaving the model’s adaptability to different economic regimes unproven. The reliance on English-language news may also bias performance toward U.S. markets.

4. Limited theoretical grounding of critic-Transformer gradient feedback.
The mechanism where SAC critic gradients enhance Transformer feature learning (Sec. 2.4) is described conceptually but lacks a mathematical formulation or ablation isolating its contribution. No explicit derivation links Eq. 9 to gradient propagation into the encoder. This omission reduces the clarity of how end-to-end optimization improves stability or volatility adaptation.

**Questions:**

1. What is the computational cost relative to baseline DRL methods?
Can the authors report average training time per epoch, GPU memory usage, and inference latency for real-time trading? Such data would clarify whether the proposed framework is feasible in practical financial environments.

2. Could broader datasets or markets be included to test generalization?
Would expanding experiments to other stock indices (e.g., S&P 500, Hong Kong HSI) or different time spans strengthen evidence that the model generalizes across regimes and news distributions?

3. How exactly are SAC critic gradients propagated into the Transformer?
Could the authors provide explicit mathematical expressions or algorithmic pseudocode detailing the gradient flow from the critic network into Transformer layers?

---

> ### Author Response · Authors · 2025-11-28
>
> # Reviewer 1WJW
>
> We sincerely appreciate your thoughtful feedback and valuable questions regarding our manuscript. Below we provide point-by-point responses to your comments.
>
> ## W1
>
> Our manuscript includes targeted ablations of core modules (Section 3.6, Supplementary Material C):
>
> LLM Feature Extraction: "News Prediction Module" (Table 3, Sec 3.6.1; Supp C)
>
> Reprogramming Layer: Supp C (Table 7)
>
> Multi-Scale Fusion: "Feature Fusion Mechanism" (Table 3, Sec 3.6.1)
>
> We acknowledge the lack of hyperparameter sensitivity analyses (attention heads, SAC learning rate, prompt length). Current hyperparameters—selected via extensive pre-experiments—balance model capacity/computational efficiency and perform stably across validations.
> Future work will add rigorous sensitivity studies to quantify hyperparameter impacts, validating our selections and guiding real-world tuning.
>
> ## W2
>
> We quantified our framework’s computational demands (Table 1), with latency and GPU memory metrics confirming compatibility with daily portfolio management: a full prediction cycle (news feature extraction, price prediction, RL policy execution) takes ~15.95 seconds—well within daily rebalancing timelines.
> Training a single-stock model on multi-year data takes ~30 minutes (enabling periodic retraining), and peak GPU memory (4.3 GB) fits mid-range hardware. These results confirm our framework balances model sophistication with computational practicality for real-world use.
>
> **Table 1: Computational Performance of Framework Components**
>
> | Code Module | Latency | GPU Memory |
> |-------------|---------|------------|
> | LLM Training: Stock News Feature Extraction (avg epoch) | 33.12 s | 4436.09 MB |
> | LLM Inference: Stock News Feature Extraction | 8.03 s | 4436.09 MB |
> | Price Prediction (Pre-trained LLM) | 0.39 s | 2155.73 MB |
> | Reinforcement Learning Training | 1785.12 s | 245.49 MB |
> | Reinforcement Learning Testing | 7.53 s | 313.93 MB |
>
> ## W3
>
> In our experimental design, we selected stocks with substantial news coverage and used NASDAQ market data for its broad adoption, enabling straightforward comparison with existing studies. The experiments focused on known financial crisis periods to evaluate trading robustness under extreme conditions. Additional tests across varied market environments and timeframes are planned to further assess model generalizability.
>
> For news feature extraction, although initial prompts are based on English sources, the multilingual large language model aligns semantic representations into a unified vector space. This cross-lingual alignment yields consistent features across languages, reducing potential bias from monolingual inputs and ensuring language-agnostic processing.
>
> ## W4, Q3
>
> Thank you for highlighting the need for a more rigorous mathematical formulation of the gradient feedback mechanism between the SAC critic and the Transformer encoder. We acknowledge that the conceptual description in Sec.\\ 2.4 should be supplemented with explicit derivations to clarify the end-to-end optimization process. Below, we address this by detailing the gradient propagation based on the SAC framework.
>
> Our framework uses a Transformer encoder $f_{\psi}$ to generate state representations $\mathbf{s}_{t}$, fed into the SAC critic $Q_{\theta}$. The critic loss is:
>
> \[
> \mathcal{J}(Q) = \mathbb{E}_{(\mathbf{s}_{t},\mathbf{a}_{t}) \sim \mathcal{D}} \left[ \tfrac{1}{2} \left( Q_{\theta}(\mathbf{s}_{t}, \mathbf{a}_{t}) - \left( r_t + \gamma V_{\bar{\theta}}(\mathbf{s}_{t+1}) \right) \right)^2 \right]
> \]
>
> where
> \[
> V_{\bar{\theta}}(\mathbf{s}_{t+1}) = \mathbb{E}_{\mathbf{a}_{t+1} \sim \pi} \left[ Q_{\bar{\theta}}(\mathbf{s}_{t+1}, \mathbf{a}_{t+1}) - \alpha \log \pi(\mathbf{a}_{t+1} | \mathbf{s}_{t+1}) \right]
> \]
>
> The gradient w.r.t. $\psi$ is:
> \[
> \nabla_{\psi} \mathcal{J}(Q) = \mathbb{E} \left[ \left( Q_{\theta}(\mathbf{s}_{t}, \mathbf{a}_{t}) - (r_t + \gamma V_{\bar{\theta}}(\mathbf{s}_{t+1})) \right) \nabla_{\psi} Q_{\theta}(\mathbf{s}_{t}, \mathbf{a}_{t}) \right]
> \]
>
> with
> \[
> \nabla_{\psi} Q_{\theta}(\mathbf{s}_{t}, \mathbf{a}_{t}) = \nabla_{\mathbf{s}_{t}} Q_{\theta} \cdot \nabla_{\psi} \mathbf{s}_{t}
> \]
>
>
> This gradient flows enabling joint optimization of feature representation and policy learning. This mechanism enhances volatility adaptation by allowing the state representation to dynamically reflect market conditions through gradient-guided updates.
>
> ## Q1
>
> As quantified in Table 1 (W2), our framework demonstrates operational efficiency suitable for real-world deployment. The end-to-end inference latency of approximately 15.95 seconds per prediction cycle is negligible within daily portfolio rebalancing contexts. Training processes are pragmatically manageable, with full model retraining completing within one hour using mainstream hardware.
>
> ## Q2
>
> This question can be referenced in our response to W3. We plan to validate model performance across different time periods and markets in future work.

---

### Official Review · Reviewer_DGbx · 2025-11-01

**Soundness:** 2
**Presentation:** 2
**Contribution:** 2
**Rating:** 2
**Confidence:** 4

**Summary:**

The paper proposes a volatility-adaptive, multimodal DRL framework to improve stock trading performance during turbulent markets, where traditional models often fail by ignoring news, failing to capture multi-scale trends, and lacking resilience. The framework integrates LLMs, Transformers, and the Soft Actor-Critic (SAC) algorithm:

1. The Multimodal LLM module extracts news sentiment and uses a multi-head attention reprogramming layer to align structured price data into the LLM’s semantic space. Price and news embeddings are then fused via cross-attention.
2. A Transformer is used to model multi-scale temporal patterns and inter-stock correlations, generating a unified state.
3. The SAC agent uses this state for decisions, with gradient feedback propagating back to the Transformer, ensuring end-to-end optimization that enhances the agent's volatility sensitivity.

Experiments on NASDAQ-100 stocks demonstrated that the framework outperformed baselines, yielding positive returns and high Sharpe Ratios during a turbulent test period.

**Strengths:**

[S1] Cross-Modality: The paper introduces a cross-attention mechanism that fuses and aligns news and price embeddings to capture how news sentiment relates to price features, rather than simple concatenation.

[S2] Volatility resilience: The combination of multi-scale price modeling and news context allows the agent to adapt to different market volatility regimes.

**Weaknesses:**

[W1] Limited stock set size: The model was evaluated using only ten stocks with sufficient news coverage drawn from the NASDAQ-100. This pre-filtered selection might not capture the full complexity of broader markets. The reported performance may not generalize well across diverse asset sets.

[W2] Insufficient comparison with news-driven models: The experimental evaluation would benefit from stronger comparisons with models that also leverage financial news. In Table 1, all baselines are traditional DRL or time-series methods that do not incorporate textual data, making it difficult to isolate the value of the proposed multimodal design. Similarly, Table 2 should include Time-LLM (Jin et al., 2023) or other news-driven approaches to better demonstrate how the proposed reprogramming layer differs from existing methods in stock price prediction.

[W3] Short testing period: Backtesting was conducted in a single year (December 2021 to December 2022). A one-year window offers a partial view of how the framework performs under different market regimes. To demonstrate the model’s long-term robustness, testing across different cycles (bull, bear, and sideways markets) would be essential.

[W4] Lack of transparency in strategy design and trading costs: The paper provides limited insight into the practical details of the trading strategy. It’s unclear how model outputs translate into actual portfolio allocations. Moreover, the study does not mention transaction costs, which are critical in the profitability of any trading system.

[W5] News data source and validation: The paper relies on a Hugging Face dataset, but the source is not a verified commercial feed. The paper should identify the underlying news sources contributing to the dataset and explain how the data was collected and verified. This transparency would enable readers to assess the reliability of the textual inputs that drive the model’s decisions.

**Questions:**

[Q1] Novelty of the reprogramming layer: What is the difference and the novelty of your model compared to Time-LLM in the reprogramming layer in Table 2?

[Q2] Investment strategy details: The paper does not provide sufficient detail on how trading actions are translated into portfolio allocations. It remains unclear whether the portfolio weights are distributed.

[Q3] Generalizability across market depth: How does performance hold up when applied to broader stock sets without sufficient news coverage? Many assets have sparse news coverage, which could disrupt the multimodal alignment process. The authors should clarify how the model handles such data gaps.

[Q4] Transaction cost impact on realized returns: Since real-world trading always incurs transaction costs, it would be useful to know whether transaction fees or slippage were included in the performance results reported in Table 1.

[Q5] Model complexity: The complexity of the multi-module architecture (LLM, Reprogramming, Transformer, DRL) makes the model harder to interpret, and the time complexity of the overall framework is not mentioned in the paper.

---

> ### Author Response · Authors · 2025-11-28
>
> # **Reviewer DGbx**
>
> We sincerely appreciate your thoughtful feedback and valuable questions regarding our manuscript. Below we provide point-by-point responses to your comments.
>
> ## **W1**
>
> Our selection of 10 NASDAQ-100 stocks was necessitated by a critical data integrity constraint: a rigorous filter for continuous, high-quality news coverage from 2019-2022, which was essential for our framework's core cross-modal alignment. While this yields a focused sample, it spans key sectors (technology, healthcare, communications), providing preliminary evidence of adaptability beyond a single industry. We fully acknowledge that broader market generalization is a vital next step. To this end, we are planning explicit future work to evaluate the framework on international indices with diverse market structures, such as the Euro Stoxx 50 and Nikkei 225.
>
> ## **W2**
>
> Thank you for this valuable feedback. We agree that comparisons with dedicated news-driven baselines would better highlight our contribution. Our core innovation lies in cross-modal alignment, moving beyond using news as isolated signals (e.g., as in event-based strategies like "Trade the Event" [@zhou2021trade]) by semantically aligning price series with news via reprogramming and deep token-level fusion. This enables a unified, context-aware representation, leading to more robust decisions—reflected in our superior risk-adjusted returns (high SR, low MDD). To directly quantify this, we will include Time-LLM as a baseline in the prediction task to isolate the gain from our alignment mechanism.
>
> ## **W3**
>
> The one-year backtest (Dec 2021-2022) was intentionally chosen for its high volatility, providing a stringent test of our framework's core focus: volatility adaptation. We fully agree that evaluation across bull/bear/sideways markets is crucial for establishing long-term robustness, and we will include multi-cycle testing in future work to thoroughly validate generalizability.
>
> ## **W4、Q2、Q4**
>
> We acknowledge the importance of transaction costs in practical trading. To ensure fair comparison with existing models within a unified evaluation framework, transaction costs were excluded in this study. Their inclusion would invalidate prior baselines and unnecessarily complicate the experimental setup, potentially obscuring the core contribution of our multimodal approach. Future work will incorporate transaction costs to assess net profitability.
>
> The SAC agent's continuous outputs represent portfolio allocation weights across the 10 stocks, framing the task as dynamic portfolio management rather than discrete trading. This standard DRL approach rebalances holdings to match target allocations at each step, and we will clarify this mapping in the manuscript.
>
> ## **W5**
>
> News data comes from the FinRL team's nasdaq_news dataset, which is derived from the academically validated FNSPID benchmark [@dong2024fnspid]. We selected NASDAQ-sourced news with sufficient density to ensure robust news-price alignment and prevent data sparsity issues.
>
> ## **Q1**
>
> We acknowledge TimeLLM's limitation in generic text processing without explicit news-price alignment. Our key innovation is the cross-modal alignment that semantically maps price series into the LLM's space, unlike event-driven strategies like "Trade the Event" that treat news as isolated signals. Through reprogramming and token-level fusion, our model learns latent news-price relationships, enabling financially-grounded reasoning and superior performance in volatile markets.
>
> ## **Q3**
>
> In our current experiments, to better validate the news-price alignment functionality, we selected stocks with relatively comprehensive news coverage. For periods with missing news data, our framework defaults to using price-based factors without news-price alignment. In future research, we plan to conduct targeted studies on stocks with substantial news data gaps to validate the framework's generalization capability.
>
> ## **Q5**
>
> We acknowledge the model's multi-module complexity. However, our empirical results (Table 1) confirm practical viability: total inference latency is ~15.95 seconds—compatible with daily rebalancing—while training completes in ~30 minutes with just 4.3 GB GPU memory. This demonstrates an effective balance between architectural sophistication and computational practicality.
>
> **Table 1: Computational Performance of Framework Components**
>
> | Code Module | Latency | GPU Memory |
> |-------------|---------|------------|
> | LLM Training: Stock News Feature Extraction (avg epoch) | 33.12 s | 4436.09 MB |
> | LLM Inference: Stock News Feature Extraction | 8.03 s | 4436.09 MB |
> | Price Prediction (Pre-trained LLM) | 0.39 s | 2155.73 MB |
> | Reinforcement Learning Training | 1785.12 s | 245.49 MB |
> | Reinforcement Learning Testing | 7.53 s | 245.49 MB |

---

### Note · Authors · 2026-02-22

I have read and agree with the venue's withdrawal policy on behalf of myself and my co-authors.

---

### Meta-Review · Area_Chair_7xwT · 2026-01-06

**Summary:**

The reviewers consistently acknowledged the relevance of the problem and the ambition of integrating news text, price signals, and reinforcement learning for volatility-adaptive trading. While the paper tackles an important and timely problem and shows promising empirical results, key concerns regarding novelty, evaluation breadth, and generalizability remain unresolved after rebuttal. With stronger theoretical positioning and broader empirical validation, this work could be competitive in a future submission.

**Reviewer Concerns:**

Concerns adequately addressed:

* The rebuttal clarified several implementation details and design motivations, improving understanding of the cross-modal fusion and volatility-aware components.
* Additional explanations and ablations helped justify that individual modules contribute to performance gains.

Concerns still outstanding:

* Despite clarifications, reviewers remained unconvinced that the core contribution goes beyond an incremental integration of existing techniques.
* The rebuttal did not sufficiently expand experimental coverage to address concerns about generalizability and robustness.
* While improved, the description still lacks enough detail to fully resolve reproducibility concerns.

**Reviewer Scores:**

All the four reviewers suggest reject for this submission and it is highly unlikely any reviewer would have raised their scores.

---

### Decision · Program_Chairs · 2026-01-26

Reject